# Deciphering Differences in Microbial Community Diversity between Clubroot-Diseased and Healthy Soils

**DOI:** 10.3390/microorganisms12020251

**Published:** 2024-01-25

**Authors:** Huajun Kang, Ali Chai, Zihan Lin, Yanxia Shi, Xuewen Xie, Lei Li, Tengfei Fan, Sheng Xiang, Jianming Xie, Baoju Li

**Affiliations:** 1College of Horticulture, Gansu Agricultural University, Lanzhou 730070, China; kanghuajun1219@163.com; 2State Key Laboratory of Vegetable Biobreeding, Institute of Vegetables and Flowers, Chinese Academy of Agricultural Sciences, Beijing 100081, China; chaiali@caas.cn (A.C.); 15528289165@163.com (Z.L.); shiyanxia@caas.cn (Y.S.); xiexuewen@caas.cn (X.X.); lilei01@caas.cn (L.L.); fantengfei@caas.cn (T.F.); xiangsheng@caas.cn (S.X.)

**Keywords:** clubroot disease, soil chemical properties, soil microbiomes, microbial community diversity, microbial network

## Abstract

Clubroot (*Plasmodiophora brassicae*) is an important soilborne disease that causes severe damage to cruciferous crops in China. This study aims to compare the differences in chemical properties and microbiomes between healthy and clubroot-diseased soils. To reveal the difference, we measured soil chemical properties and microbial communities by sequencing 18S and 16S rRNA amplicons. The available potassium in the diseased soils was higher than in the healthy soils. The fungal diversity in the healthy soils was significantly higher than in the diseased soils. Ascomycota and Proteobacteria were the most dominant fungal phylum and bacteria phylum in all soil samples, respectively. Plant-beneficial microorganisms, such as *Chaetomium* and *Sphingomonas*, were more abundant in the healthy soils than in the diseased soils. Co-occurrence network analysis found that the healthy soil networks were more complex and stable than the diseased soils. The link number, network density, and clustering coefficient of the healthy soil networks were higher than those of the diseased soil networks. Our results indicate that the microbial community diversity and network structure of the clubroot-diseased soils were different from those of the healthy soils. This study is of great significance in exploring the biological control strategies of clubroot disease.

## 1. Introduction

Clubroot, caused by the obligate biotrophic soilborne pathogen *Plasmodiophora brassicae*, is considered a devastating disease of *Brassica* crops [1]. At present, clubroot is widespread in China and is becoming very problematic. The pathogen can survive in the soil in the form of resting spores for decades, and the number of spores in the soil increases rapidly in the continuous monoculture of *Brassica* crops [2]. Many control measures have been used for the management of clubroot, including the deployment of resistant cultivars, soil fumigation, and the application of fungicides [3]. However, successful management of the disease is difficult because of monetary and labor costs.

It is known that the occurrence of soilborne diseases correlates with soil properties, including the soil microbiome. The microorganisms in the rhizosphere are extremely abundant and are the first line of defense against pathogen invasion [4]. The destruction of this microbial defense can cause plants to be attacked by harmful microbes [5]. In addition, plants can recruit beneficial microbes to suppress pathogen invasion through the secretion of exudates [6]. Therefore, soil microorganisms play a vital role in regulating plant health and growth [7]. Using microorganisms to control soilborne diseases is considered a desirable approach. Microorganisms can protect plants from pathogens through many mechanisms, such as competing with pathogens for nutrients, producing antagonists that inhibit pathogens, or eliciting plant systemic defense [8,9]. However, the successful colonization of bioinoculants in plant rhizosphere depends on many factors, such as the composition and interactions with and within the locally adapted resident microbiota and soil attributes [10,11]. Therefore, a better knowledge of soil microbial community is essential for the prevention and control of plant diseases.

Many studies have suggested that the soil microbial community composition, diversity, structure, and function are related to the outbreak of plant soilborne diseases. For instance, a decrease in Firmicutes and Actinobacteria abundance in the tomato rhizosphere led to the outbreak of bacterial wilt disease [12]. Soil microbial communities containing high abundances of beneficial microorganisms can more effectively inhibit pathogens to maintain plant health [13]. Wei et al. [14] found that the initial soil microbiome composition and functioning can predetermine future plant health. High microbial diversity is likely related to the complexity of microbial interactions and, therefore, increases soil fungistasis and resistance to microbial invasion [15,16]. Previous studies have demonstrated that the disease severity of clubroot is associated with soil biotic factors [17]. However, knowledge of the differences in soil microbes between healthy and clubroot-diseased soils is limited.

Soil chemical properties also affect plant health and pathogen survival in soils [18,19]. Across a variety of crops, soil pH, organic matter content, extractable calcium, and boron contents are negatively correlated with disease severity [20]. In addition, the soil’s chemical properties can also indirectly influence the expression of soilborne diseases by affecting soil microbial communities [21]. For instance, soil acidification can largely impact bacterial communities and further reduce the capacity of soils to defend against fungal pathogens [22]. It has been reported that the occurrence of clubroot disease is negatively correlated with soil pH, boron, and calcium levels but positively corrected with bulk density and soil humidity [23,24,25,26]. However, different studies often show inconsistent results [27]. Hence, it is necessary to clarify the correlation between soil properties and plant health.

The present study aims to evaluate the characteristics and differences between clubroot-diseased and healthy soils, including the soil’s chemical properties and microbial properties. We hypothesized that (1) soil chemical properties and microbial community are correlated with clubroot disease and (2) that the soil microbial community varies in healthy and clubroot-diseased soils. We employed Illumina-MiSeq sequencing to compare the rhizosphere soil microbial communities of healthy and clubroot-diseased soils from different regions. As a result, this study provides a foundation for the biological control of clubroot disease in cruciferous crops.

## 2. Materials and Methods

### 2.1. Site Description and Sample Collection

All soil samples were collected from cruciferous vegetable fields designated as Licang (120.4267° E, 36.1511° N), Shandong Province; Youxian (108.6638° E, 30.3212° N), Sichuan Province; and Lichuan (109.7990° E, 30.7344° N) and Jianshi (31.5959° N, 104.8624° E), Hubei Province, China (Appendix A). The soils covered two different soil types (Cambisols and Luvisols), and other detailed information is listed in Table 1. Each field has a high clubroot disease incidence (over 50% over the last five years).

For each field, 3 random subplots (approximately 60 m^2^) were chosen, and soil samples from approximately 10 healthy (healthy soil) or 10 clubroot-diseased (diseased soil) plants from each subplot were collected using the checkerboard sampling method in August 2020. Briefly, each subplot was divided into 10 areas, and the rhizosphere soils of healthy and clubroot-diseased plants in the central point of each area were collected via manual shaking [28]. In each field, not only did the healthy plants have no root gall, but the aboveground plant growth was also good and consistent. Accordingly, the diseased plants showed noticeable wilting in the aboveground parts, and the root systems showed typical gall formation. The 10 healthy or 10 diseased soils were mixed to form one composite sample. The composite samples were placed into separate sterile bags and transported to the laboratory on ice. Each composite sample was ground and sieved through a 2 mm sieve and then divided into two subsamples: one portion was used for chemical property analysis after air-drying, and another was stored at −80 °C for subsequent DNA extraction.

### 2.2. Quantification of P. brassicae

For each composite soil, total DNA was extracted from 0.5 g of soil using the FastDNA^®^ Spin kit (MO BIO Laboratories, Inc., Carlsbad, CA, USA), according to the manufacturer’s protocol. Each composite soil sample was extracted in triplicate, and the extracted DNA solutions were pooled. We quantified *P. brassicae* using quantitative polymerase chain reaction (qPCR) according to the methods described by Chai et al. [29]. Fluorescence was detected after each cycle and the Log of the resting spore numbers was calculated according to the method described in our previous work [29].

### 2.3. Soil Chemical Properties

The soil pH was determined using a pH meter (Shanghai Sanxin Instrumentation, Inc., Shanghai, China) in a 1:5 (m/m) soil:water solution. Soil organic matter (SOM) was determined using the potassium dichromate external heating method [30]. The total nitrogen (TN) level was determined using the semi-micro Kjeldahl method. Available nitrogen (AN) was determined using the alkaline-hydrolyzable diffusion method [31]. Available phosphorus (AP) was determined using a protocol according to Shen et al. [32]. Available potassium (AK) was extracted with ammonium acetate and determined via flame photometry using an FP640 Flame Photometer (Shanghai Instruments Group Co., Ltd., Shanghai, China). The cation exchange capacity (CEC) was measured according to the method of Mulvaney et al. [33]. Electrical conductivity (EC) was determined using a conductivity meter (Shanghai Leici Instrument Factory, Shanghai, China) in a soil water suspension (1:5 *w*/*v*) after being shaken at 200 r min^−1^ for 15 min at room temperature. The exchangeable calcium (Ca) and available boron (B) were measured using the methods of Bao [34] and Bhering et al. [35], respectively.

### 2.4. Amplicon Sequencing and Data Processing

Soil total DNA was extracted as described above. The primer set ITS1F (5′-CTTGGTCATTTAGAGGAAGTAA-3′) [36] and ITS2 (5′-GCTGCGTTCTTC ATCGATGC-3′) were selected to target the fungal ITS1 region. 338F (5′-ACTCCTACGGGAGGCAGCAG-3′) and 806R (5′-GGACTACNNGGGTATC TAAT-3′) [37] were used to amplify the V4 hypervariable regions of the bacterial 16S rRNA gene. Amplicons were sequenced with the MiSeq platform at Allwegene CO., LIMITED (Beijing, China).

The raw data were first screened, and low-quality sequences with a quality score lower than 20 or a length shorter than 230 bp were discarded. Sequences with a 97% identity level were assigned to the same OTU using the Uparse algorithm in the Vsearch (v2.7.1) software. The chimeras were removed using Usearch (version 8.0.1623). The taxonomy of each 16S rRNA and ITS gene sequence was classified using the SILVA database and the UNITE database, respectively, with a confidence threshold of 80%. The alpha diversity indices (Chao1 index and Shannon index) were calculated using the QIIME software (v1.8.0). All raw sequences were uploaded to the NCBI Sequence Read Archive (SRA) database under the accession number PRJNA982587.

### 2.5. Network Analysis

To determine the complexity of the interactions between microbial taxa, co-occurrence networks were constructed using the rhizosphere microorganisms based on a Spearman correlation. In this study, the 200 most abundant fungal and bacterial OTUs from healthy and diseased soil samples were used for the network constructions (*n* = 12). The Benjamini–Hochberg FDR controlling procedure was used to correct the *p*-value via multiple tests. Correlation data were filtered with a cut-off at an absolute r-value of 0.6–0.93 and a *p*-value < 0.05. Nodes in the network represent gate-level classification units, and line-connecting nodes represent significant positive or negative correlations. The clustering coefficient (the clustering coefficient shows how well a node is connected with the neighbor nodes) and network density (the ratio of realized to possible edge numbers) were chosen to reflect the changes in soil microbiome associations [38]. Gephi (version 0.9.2) was used to visualize the co-occurrence networks.

The topological roles of nodes were determined using among-module connectivity (Pi) and within-module connectivity (Zi). All the nodes were classified into four categories, including module hubs (named generalists, Pi ≤ 0.62 and Zi > 2.5), network hubs (named supergeneralists, Zi > 2.5 and Pi > 0.62), connectors (also named generalists, Pi > 0.62 and Zi ≤ 2.5), and peripherals (named specialists, Pi ≤ 0.62 and Zi ≤ 2.5). Module hubs and connectors are considered putative keystone taxa [39,40]. The connectors, module hubs, and network hubs are core species that play important roles in maintaining network stability [41]. The topology property parameters of the network were calculated using the R (3.6.0) packages “Hmisc” and “igraph”. Zi–Pi plots were visualized using ggplot2 in the R platform [42].

### 2.6. Statistics Analysis

Differences in *P. brassicae* abundance, soil properties, and fungal and bacterial alpha diversity indices between the healthy and diseased soils were compared via one-way ANOVA (analysis of variance) followed by Tukey’s test (*p* < 0.05). The Wilcoxon rank-sum test was performed to identify differences in bacterial and fungal taxa between the healthy and diseased soils [43]. Pearson’s correlations between chemical parameters and disease incidence and between the abundance of *P. brassicae* and the abundant taxa were also performed. All the above statistical analyses were performed using the SPSS 22.0 software (SPSS Inc., Chicago, IL, USA). To compare microbial community structures across all samples, a hierarchical cluster tree was constructed using the unweighted paired-group method with arithmetic means (UPGMA) based on the Bray–Curtis distance matrix.

## 3. Results

### 3.1. P. brassicae Abundance in Different Fields

The abundance of *P. brassicae* was analyzed in all soil samples collected from four regions (Figure 1). The results showed that the population of *P. brassicae* ranged from 3.13 Log CFU/g to 8.04 Log CFU/g in the soil samples. Quantitative analysis showed that the colonization of *P. brassicae* was less than 3.92 Log CFU/g in the healthy soil samples from the four different fields, but more than 5.28 Log CFU/g of *P. brassicae* was detected in the diseased soils. The *P. brassicae* population in the healthy soil samples collected from Licang (3.69 Log CFU/g), Lichuan (3.92 Log CFU/g), and Jianshi (3.80 Log CFU/g) was significantly (Tukey’s test, *p* < 0.05) higher than that collected from Youxian (3.13 Log CFU/g). However, the *P. brassicae* population in the diseased soil samples collected from Youxian (8.04 Log CFU/g) was significantly higher than those collected from the other three fields. Among the four diseased soil samples, the lowest level (5.28 Log CFU/g) of resting spore infestation was detected in Lichuan. Within the same soil type, the abundance of *P. brassicae* in the healthy soils was significantly lower than that of the diseased soils. For the same field, the *P. brassicae* population in the diseased soil samples was significantly (Tukey’s test, *p* < 0.05) higher than in the healthy soil samples.

### 3.2. Soil Chemical Properties

The chemical properties of the soils are presented in Table 2. The pH, AN, AP, AK, and Ca contents in healthy soil samples were significantly (Tukey’s test, *p* < 0.05) different between the four regions. For diseased soil samples, AK, AP, and Ca levels in the four regions were significantly different. For the same field or each of the two soil types, compared with the healthy soils, the diseased soils had higher AK content but with no significant difference. There were significant differences in Ca levels between the healthy and diseased soils in the same field or each soil type. The pH values of the healthy and diseased soils were significantly different in Youxian, Lichuan, and Licang but did not show a consistent tendency in different fields. The CEC and B contents showed no difference between the healthy and diseased soils in the same field. However, other chemical properties (SOM, TN, AN, AP, and EC) showed different trends between the healthy and diseased soils in different regions.

### 3.3. Microbial Community Diversity

Altogether, 3, 685, 142 ITS sequence reads and 1, 642, 542 16s RNA effective sequence reads were obtained across the 24 soil samples. In total, 3652 fungal OTUs and 9185 bacterial OTUs were identified from these reads. We used the OTU number, Chao 1 (richness) and Shannon indices to evaluate and compare the diversity and richness of microbial communities among different soil samples (Table 3). For each field, there was no significant difference in the number of fungal OTUs and richness between the healthy and diseased soils. The fungal Shannon indices were higher in the healthy soils than in the diseased soils but with no significant difference except Licang. For bacteria, the OTU number and Shannon indices of the healthy and diseased soils were significantly (Tukey’s test, *p* < 0.05) different in Licang, Youxian, and Lichuan. The Chao1 indices of the healthy and diseased soils were significantly different in Youxian and Lichuan. The results indicated that the overall effect of the diseased soils on microbial diversity and richness was not significant. The fungal Shannon diversity indices in the healthy soil groups were significantly (Tukey’s test, *p* < 0.05) higher than those in the diseased soil groups (Table 4). However, there was no significant difference in the fungal OTU number and Chao1 indices between the healthy soil and diseased soil groups. For bacteria, there was no significant difference in OTU number, richness, and Shannon diversity indices between the healthy soil and diseased soil groups.

Hierarchical cluster analysis revealed that the same field presented similar fungal and bacterial microbial community structures (Figure 2). The fungal and bacterial communities from the healthy soils were separated from the diseased soils collected from the same field site, suggesting contrasting microbial community structures according to disease status.

### 3.4. Microbial Composition

A total of 16 fungal phyla were identified from all of the soil samples. Ascomycota, Mortierellomycota, Basidiomycota, and Chytridiomycota were the dominant phyla in all the samples, with Ascomycota being the most dominant phylum, accounting for 82.83% (Licang), 61.11% (Youxian), 64.33% (Lichuan), and 51.63% (Jianshi) in the healthy soils and 87.49% (Licang), 55.08% (Youxian), 84.34% (Lichuan), and 64.37% (Jianshi) in the diseased soils (Figure 3A). For the bacterial community, all OTUs were classified into 45 phyla. Proteobacteria, Acidobacteriota, Actinobacteriota, Chloroflexi, Gemmatimonadota, Bacteroidota, and Patescibacteria were the dominant phyla across all the samples (Figure 3B). The most abundant phylum in soils was Proteobacteria, which accounted for 28.21% (Licang), 35.21% (Youxian), 47.56% (Lichuan), and 37.08% (Jianshi) in the healthy soils and 38.05% (Licang), 36.41% (Youxian), 39.10% (Lichuan), and 31.08% (Jianshi) in the diseased soils. Among the abundant phyla (relative abundance > 0.1%), a higher abundance of Gemmatimonadota was in the healthy soil samples compared with the diseased soil samples within the same field. However, the other abundant phyla did not show a consistent tendency between the healthy and diseased soils in the same field. There was no significant difference in the fungi and bacteria at the phylum level between the healthy and diseased soils.

We further analyzed the association of the 15 most abundant fungal and bacterial genera with the healthy and diseased soils (Figure 4). The abundance of *Chaetomium* and *Botryotrichum* in the healthy soils was significantly higher than in the diseased soils (Wilcoxon rank-sum test, *p* < 0.05), but other genera showed no obvious difference between the diseased and healthy soil samples. For bacteria, *Sphingomonas* was more abundant in the healthy soils than in the diseased soils. There was no significant difference in the abundance of other bacterial genera between the healthy and diseased soils.

### 3.5. Effects of Soil Variables and Microbial Species on Clubroot Disease

Relationships between chemical properties and disease incidence and between the abundance of *P. brassicae* and the 15 most abundant taxa were analyzed using Pearson’s correlation coefficient analysis. Disease incidence was only significantly positively correlated with CEC (Table 5). There was no correlation between disease incidence and other chemical properties. For fungal genera, the abundance of *P. brassicae* was significantly negatively correlated with *Chaetomium*, *Botryotrichum*, and *Acremonium*. With to bacteria, only *Bacillus* was significantly positively correlated with the abundance of *P. brassicae* (Table 6).

### 3.6. Network Analysis

To investigate the relationship between the soil microbe–microbe interactions and clubroot disease, we performed a microbial co-occurrence network analysis (Figure 5). For fungal networks, the healthy soils (3657 links) had higher link numbers than the diseased soils (2980 links) (Table 7). Both the healthy and diseased soils had higher positive link numbers than negative link numbers. The positive link/negative link ratio in diseased soils (3.02) was higher than that in healthy soils (2.15), demonstrating that the diseased soils contained more positive co-occurrence relationships than the healthy soils. The clustering coefficient of the healthy soils (0.57) was higher than that of the diseased soils (0.53). The network density of microbiome networks in the healthy soils was 0.19 higher than 0.15 in the diseased soils. The modularity value of the healthy (0.39) and diseased (0.40) soils was similar. In the fungal networks, 22 fungal nodes out of the top 200 nodes were sunk into “connectors” (Figure 6A and Appendix A). Members from Ascomycota, Basidiomycota, Chytridiomycota, and Glomeromycota were identified as keystone fungal taxa. No network hubs and module hubs were found in the fungal networks.

For the bacteria networks of the healthy and diseased soils, the number of links was 8789 and 6750, respectively. The diseased soils (1.33) had a higher positive link/negative link ratio than the healthy soils (1.16). The network density and the clustering coefficient in networks of the healthy and diseased soils were 0.44 and 0.34 and 0.75 and 0.68, respectively. The healthy (0.31) soils had higher modularity than the diseased (0.18) soils. We found that 13% of the top 200 nodes were connectors (26 nodes), while 87% were peripherals (Figure 6B and Appendix A). Similar to the fungal networks, there were no network hubs or module hubs detected in the bacterial networks. Members from Proteobacteria, Acidobacteriota, Actinobacteriota, Bacteroidota, Chloroflexi, Myxococcota, Nitrospirota, and Verrucomicrobiota were identified as keystone bacterial taxa.

## 4. Discussion

Plant health is closely related to soil properties, including physical, chemical, and microbial attributes. Characterizing the differences in soil properties between healthy and diseased soils is an important first step to understanding the pathogenesis of soilborne disease. In this study, we compared the soil chemical properties and microbial communities of healthy and clubroot-diseased Cruciferae crops in four regions of China. We found that the fungal diversity in the healthy soils was higher than in the diseased soils, and the networks of healthy soils were more complex than those of clubroot-diseased soils.

Clubroot severity in field conditions has been suggested to be related to soil pH value [44,45]. The soil pH in the clubroot-diseased cruciferous crops ranged from 5.13 to 7.54 across the four regions. This result is in line with previous findings showing that severe clubroot developed at pH 5.0–8.0 in field conditions [46]. The soil pH did not show a consistent tendency between healthy and diseased soils in four different regions, suggesting that pH may not be an important factor affecting clubroot severity in this study. Similarly, an assessment of soil pH in commercial canola fields infested with clubroot showed that there was only a weak correlation between soil pH and clubroot level [46]. It is noteworthy that the soil pH values of Lichuan and Jianshi were lower than those of Licang and Youxian, while the corresponding bacterial Chao1 indices of healthy and diseased soils in the two regions were, overall, at low levels among the four regions. Thus, soil pH may be one of the factors affecting bacterial diversity, which is in line with previous studies [47,48]. A high calcium concentration can reduce the rate of maturation of *P. brassicae* in root hairs and clubroot severity in *Brassica* crops [49,50]. In this study, the calcium levels of the healthy and diseased soils were significantly different in the same region, but the changing trend was inconsistent in different regions. Our results indicated that there was no significant correlation between calcium levels and healthy plant status, which is in agreement with previous findings [45,51]. Previous studies have shown that boron reduces the development of clubroot in *Brassica* crops. Nevertheless, the boron levels were not significantly different for the healthy and diseased soils in the same region. This may be because other conditions were conducive to infection. At present, there are few reports on the variation in CEC in soil during the infection process of *P. brassicae* in cruciferous crops. In the present study, CEC was significantly positively related to disease incidence. Our results may indicate that CEC could be an indicator of the severity of clubroot disease.

The results showed that the microbial diversity of the healthy and diseased soils differed between regions. Some areas were the same, while others showed different microbial diversity. The Chao1 indices of the fungi were not significantly different for the healthy and diseased soils in all four regions, which is in line with previous findings [52,53]. For bacteria, the Chao1 indices of the healthy and diseased soils were significantly different in Youxian and Lichuan but not significantly different in Licang and Jianshi. The results indicated that the diseased soils in Youxian and Lichuan may have a significant effect on bacterial diversity. Taken together, the field site location had a large effect in determining microbial community diversity in the present study, which is similar to previous studies that illustrated the importance of spatial influences [47,54,55].

The composition and structure of soil microbial communities have an important impact on soil and plant health [7,56]. In the present study, the same region presented a similar microbial composition, but there were differences between different sites. For instance, Ascomycota and Basidiomycota were the most abundant fungal phyla identified in Licang, while Ascomycota and Mortierellomycota were the two most prevalent fungal phyla in Youxian, Lichuan, and Jianshi. Gemmatimonadota was relatively more abundant in the healthy soils compared with the diseased soils in each of the four regions, which is in agreement with previous results [57,58,59]. Thus, Gemmatimonadota may be an indicator of the soil supporting plant health. A recent investigation reported that members of Gemmatimonadota can encode diverse polyketide and nonribosomal peptide biosynthetic gene clusters [60]. Non-ribosomal peptides are one of the largest groups of natural microbial secondary metabolites, which are important for the producer in terms of defense responses, chemical communication, and nutrient acquisition [61,62]. The microbial community structure collected from the same region was more similar when compared with different regions, which is in agreement with previous studies [53,55].

We compared the microbial differences between groups for the 15 most abundant genera from different regions. The results showed that the relative abundance of *Chaetomium* in the healthy soils was significantly higher than in the diseased soils. Previous studies have shown that some species of *Chaetomium* can inhibit the plant pathogen through competition, mycoparasitism, antibiosis, and a combination of various mechanisms [63,64,65]. This taxon may serve as an important indicator of clubroot disease suppression in the cruciferous crop system. For bacteria, the relative abundance of *Sphingomonas* in the healthy soils was significantly higher than in the diseased soils, which is in line with previous studies [66]. *Sphingomonas* species play an important role in plant growth promotion, abiotic stress tolerance, and environmental remediation [67,68,69,70,71]. In a recent study, *Sphingomonas* was identified as one of the keystone genera associated with pathogen suppression in healthy plant microbiomes [14]. The relative abundance of *Chaetomium*, *Botryotrichum*, and *Acremonium* was significantly negatively correlated with that of *P. brassicae* (Table 6). We speculate that these species may play an important role in reducing the development of clubroot disease.

Many interactions between different microbial species in soil microbial communities formed a complex microbial network, which is vital to ecosystem stability [72,73,74]. In the present study, the co-occurrence networks revealed different patterns within the microbial communities of the healthy and clubroot-diseased soils. The healthy soils showed a higher number of co-occurrence relationships than the clubroot-diseased soils for both fungal and bacterial networks. A higher number of links means a more complex network, suggesting that the microbial networks of the healthy soils were more complex than those of the diseased soils, which followed the results of Xiong et al. [13] but contrasted with the results of Hu et al. [8]. This may be because the healthy soils had higher diversity than the clubroot-diseased soils [75]. Strikingly, the number of positive links was higher than that of the negative links for fungal and bacterial networks in the healthy and clubroot-diseased soils. More positive interactions may suggest more cooperation in soil microbial community ecosystems, which may be linked to a higher community function [76,77]. In addition, the healthy soils exhibited higher complexity, reflected by increased network density and clustering coefficients [78]. Overall, the healthy networks were more complex and stable, therefore making them better able to exclude the pathogen *P. brassicae* from the plant environment.

Connectors are essential microorganisms that act as regulators, mediators, and adaptors in a network [79]. The members of Ascomycota and Proteobacteria accounted for the largest proportion of connectors in the fungal and bacterial networks, respectively, which may be related to their high relative abundance in the soils. It is worth noting that species of *Acremonium* have been investigated as potential antagonistic agents against clubroot disease (Appendix A) [80]. Future studies would be required to investigate the control effect of the species on clubroot disease. Among connectors in the bacterial networks, some beneficial microbes, such as *Alcaligenes*, *Klebsiella*, *Nitrosospira*, and *Nitrospira*, were identified, which function in the nitrogen cycle (Appendix A) [81,82,83].

In conclusion, the bacterial and fungal communities of healthy and clubroot-diseased soils were different. The fungal diversity in the healthy soils was significantly higher than in the diseased soils. The healthy soils exhibited significantly higher relative abundances of *Chaetomium* and *Sphingomonas* than the diseased soils. The healthy soils had a higher link number, network density, and clustering coefficient for both fungal and bacterial networks and, thus, were more complex and stable than the diseased network. In addition, healthy soils had higher AK content than diseased soils. Together, these findings provide some clues into potential methods of clubroot disease management through the manipulation of microbial communities.

## Figures and Tables

**Figure 1 microorganisms-12-00251-f001:**
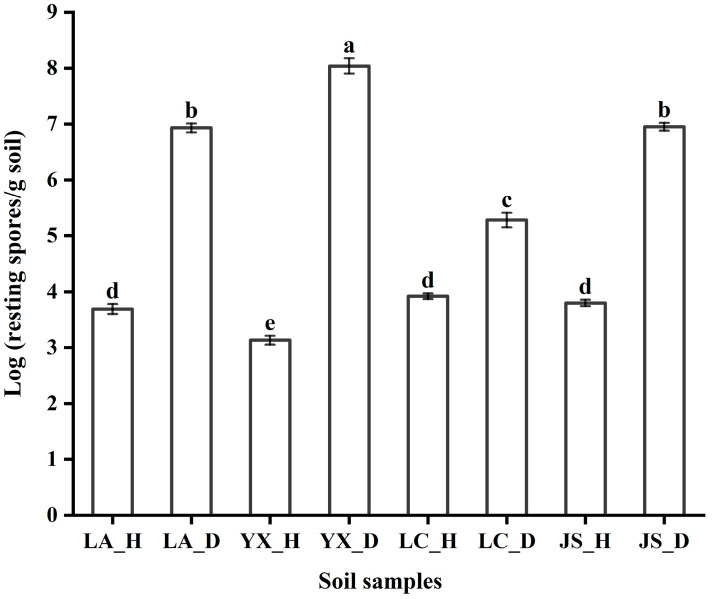
Quantification of the resting spores of *P. brassicae* in different soil samples. LA_H, YX_H, LC_H, and JS_H represent the healthy soil samples collected from Licang, Youxian, Lichuan, and Jianshi, respectively. LA_D, YX_D, LC_D, and JS_D represent the clubroot-diseased soil samples collected from Licang, Youxian, Lichuan, and Jianshi, respectively. The error bars indicate the standard deviation of the means (*n* = 3). Different letters above the bars indicate statistically significant differences according to Tukey’s test (*p* < 0.05).

**Figure 2 microorganisms-12-00251-f002:**
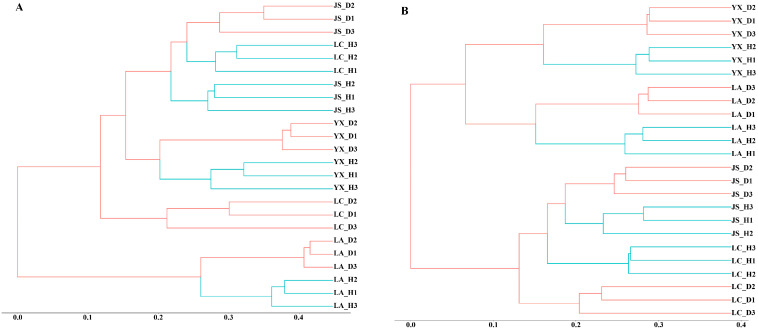
Hierarchical cluster trees constructed based on the Bray–Curtis distance matrix for the healthy and diseased soil samples. (**A**) Hierarchical cluster tree of fungi. (**B**) Hierarchical cluster tree of bacteria. LA_H, YX_H, LC_H, and JS_H represent the healthy soil samples collected from Licang, Youxian, Lichuan, and Jianshi, respectively. LA_D, YX_D, LC_D, and JS_D represent the clubroot-diseased soil samples collected from Licang, Youxian, Lichuan, and Jianshi, respectively. Green lines represent healthy soil samples and red lines represent diseased soil samples.

**Figure 3 microorganisms-12-00251-f003:**
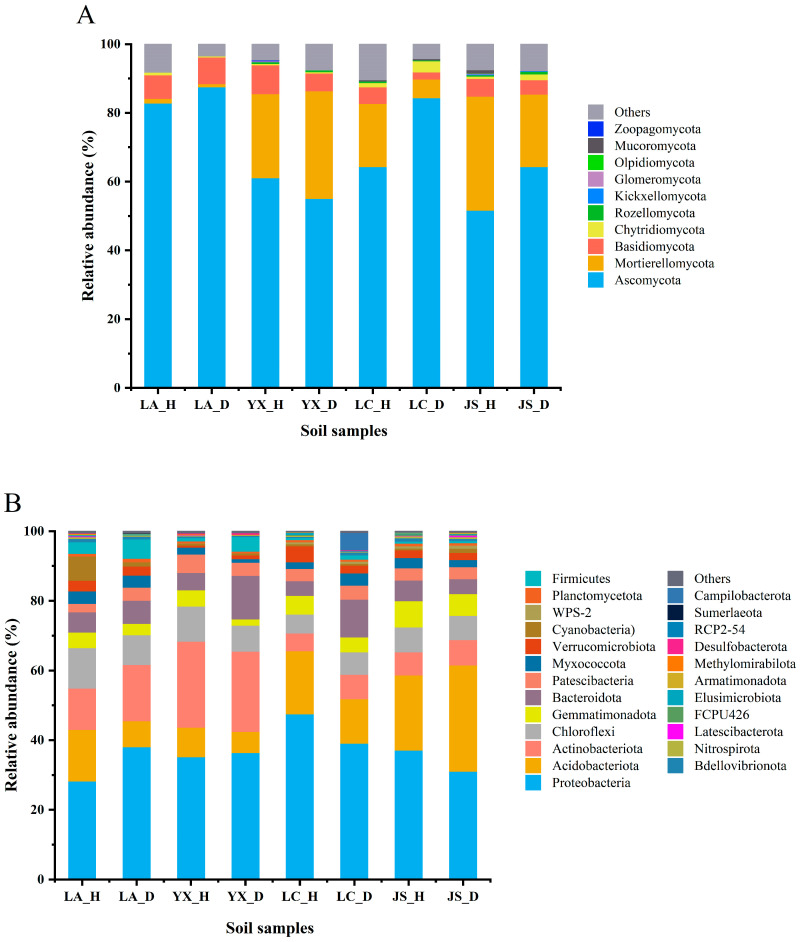
Average relative abundance of fungal (**A**) and bacterial (**B**) phyla in soil samples. LA_H, YX_H, LC_H, and JS_H represent the healthy soil samples collected from Licang, Youxian, Lichuan, and Jianshi, respectively. LA_D, YX_D, LC_D, and JS_D represent the clubroot-diseased soil samples collected from Licang, Youxian, Lichuan, and Jianshi, respectively. Phyla below 0.1% of relative abundance and the unclassified phyla are classified as “Others”.

**Figure 4 microorganisms-12-00251-f004:**
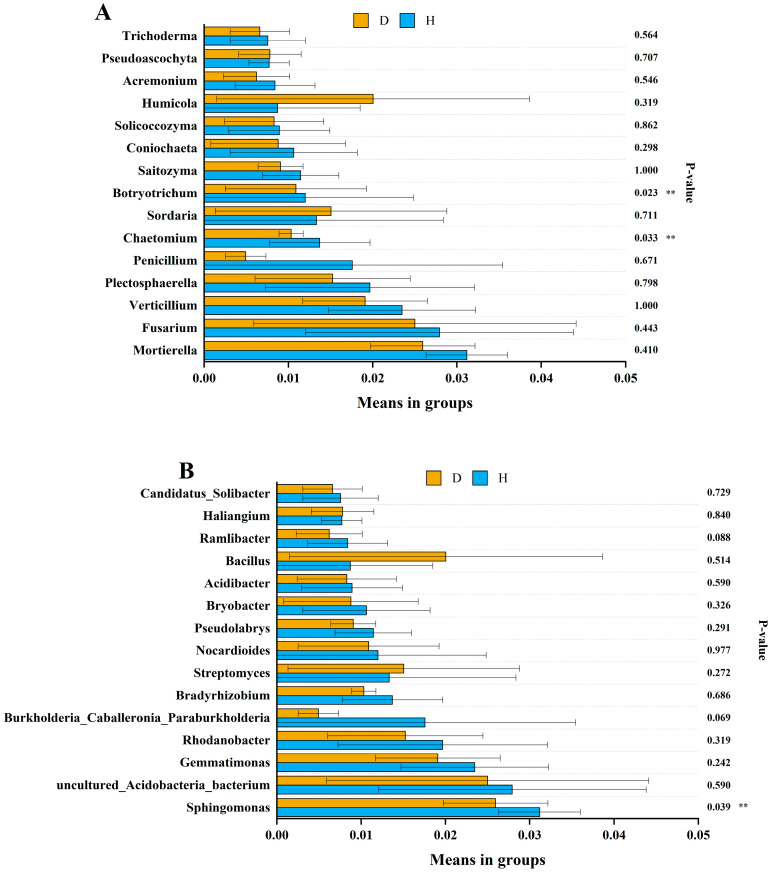
Analysis of abundance differences between the healthy and clubroot-diseased soils at the fungal (**A**) and bacterial (**B**) genus levels. D represents the diseased soils. H represents the healthy soils. The error bars indicate the standard deviation of three replicates (*n* = 12), and double asterisks indicate statistical significance (Wilcoxon rank-sum test, *p* < 0.05).

**Figure 5 microorganisms-12-00251-f005:**
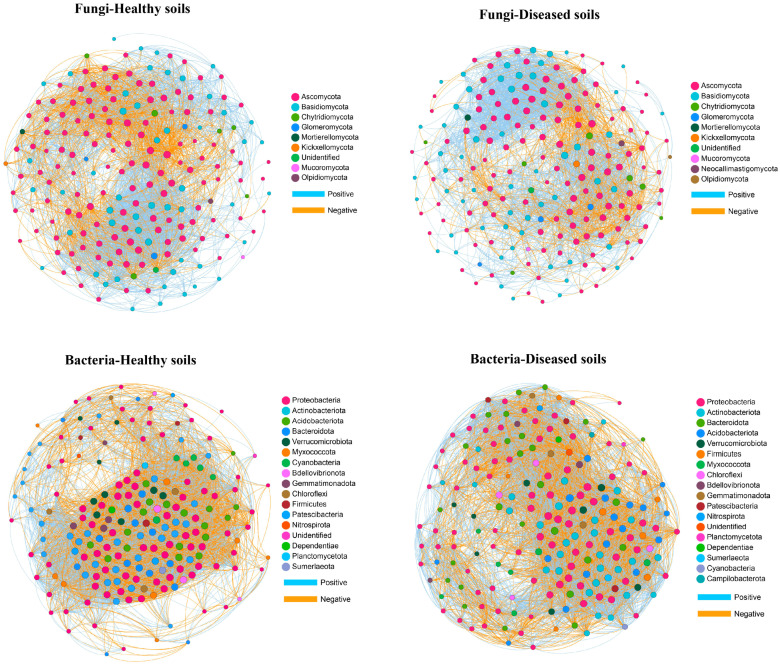
Co-occurrence networks of the 200 most abundant fungal and bacterial genera. The size of the node is equivalent to its relative abundance. Blue and yellow edges represent positive and negative interactions, respectively.

**Figure 6 microorganisms-12-00251-f006:**
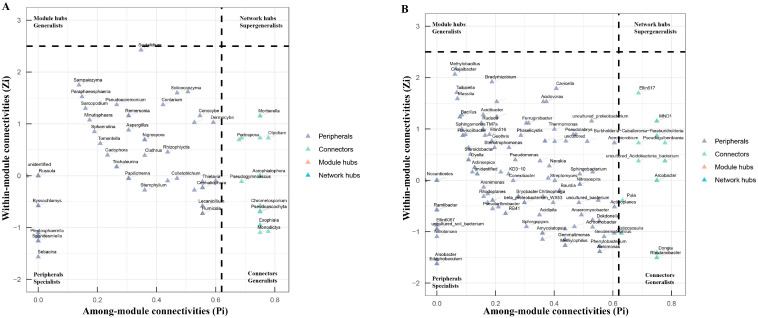
Topological roles of nodes in fungal (**A**) and bacterial (**B**) networks. The topological roles of nodes were determined using among-module connectivity (Pi) and within-module connectivity (Zi). All the nodes were classified into four categories, including module hubs (named generalists, Pi ≤ 0.62 and Zi > 2.5), network hubs (named supergeneralists, Zi > 2.5 and Pi > 0.62), connectors (also named generalists, Pi > 0.62 and Zi ≤ 2.5), and peripherals (named specialists, Pi ≤ 0.62 and Zi ≤ 2.5).

**Table 1 microorganisms-12-00251-t001:** Description of the soil samples.

Sample Location	Clubroot Infection, %	Coordinates	Altitude (m)	Mean Annual Temperature (°C)	Mean Annual Precipitation (mm)	Year	Soil Type	Crops
Licang, Shandong Province	55.38%	36.1511° N 120.4267° E	99	14.5	732.0	2020	Luvisols	Chinese cabbage
Lichuan, Hubei Province	77.46%	30.3212° N 108.6638° E	1595	12.3	1200.0	2020	Cambisols	Chinese cabbage
Jianshi, Hubei Province	72.29%	30.7344° N 109.7990° E	726	16.0	1480.0	2020	Cambisols	Chinese cabbage
Youxian, Sichuan Province	85.73%	31.5959° N 104.8624° E	568	16.4	969.6	2020	Cambisols	Cabbage

**Table 2 microorganisms-12-00251-t002:** Chemical properties of the soil samples.

Sample Location	Soil Trait	pH	SOM (g/kg)	TN (g/kg)	AN (mg/kg)	AP (mg/kg)	AK (mg/kg)	CEC (col (+)/kg)	EC (ms/cm)	Ca (g/kg)	B (mg/kg)
Licang, Shandong Province	Healthy	7.52 ± 0.01 a	19.97 ± 0.94 c	0.89 ± 0.01 d	94.73 ± 2.38 e	99.20 ± 0.42 d	113.83 ± 0.47 f	6.63 ± 0.41 d	233.67 ± 5.44 c	2.57 ± 0.04 c	0.31 ± 0.04 ab
	Diseased	7.54 ± 0.00 a	15.56 ± 0.79 d	0.82 ± 0.00 d	102.20 ± 1.14 e	97.61 ± 0.54 d	120.83 ± 0.47 f	6.57 ± 0.25 d	330.67 ± 17.25 b	2.36 ± 0.03 d	0.34 ± 0.02 ab
Youxian, Sichuan Province	Healthy	7.25 ± 0.02 b	22.55 ± 0.30 bc	1.44 ± 0.03 c	144.67 ± 0.66 d	48.10 ± 0.56 e	139.17 ± 4.92 e	12.97 ± 0.25 ab	66.53 ± 2.99 d	4.27 ± 0.04 a	0.41 ± 0.03 a
	Diseased	6.11 ± 0.04 c	24.15 ± 0.38 b	1.62 ± 0.03 b	203.93 ± 0.66 c	44.22 ± 0.55 e	247.50 ± 2.83 d	14.17 ± 0.34 a	43.37 ± 4.29 de	3.90 ± 0.02 b	0.39 ± 0.05 a
Lichuan, Hubei Province	Healthy	5.03 ± 0.05 f	35.87 ± 0.04 a	2.29 ± 0.01 a	245.00 ± 1.14 b	153.05 ± 1.80 a	389.50 ± 3.27 b	12.77 ± 0.41 ab	54.35 ± 10.30 de	1.53 ± 0.02 h	0.22 ± 0.01 c
	Diseased	5.20 ± 0.03 e	37.72 ± 0.93 a	2.31 ± 0.04 a	257.60 ± 1.14 ab	119.28 ± 0.85 c	533.17 ± 7.76 a	12.50 ± 1.02 b	66.36 ± 3.16 d	2.26 ± 0.02 e	0.25 ± 0.02 bc
Jianshi, Hubei Province	Healthy	5.50 ± 0.01 d	36.50 ± 1.45 a	2.27 ± 0.04 a	260.40 ± 2.29 a	143.63 ± 2.75 b	333.83 ± 2.62 c	10.77 ± 0.09 c	761.00 ± 25.15 a	1.68 ± 0.02 g	0.34 ± 0.02 ab
	Diseased	5.13 ± 0.02 e	36.64 ± 1.19 a	2.23 ± 0.02 a	259.93 ± 9.86 a	145.19 ± 0.58 b	340.83 ± 2.05 c	12.30 ± 0.33 bc	25.20 ± 1.26 e	1.82 ± 0.02 f	0.35 ± 0.04 ab

All data are presented as the mean ± standard deviation (*n* = 3). Values followed by different letters indicate statistically significant differences according to Tukey’s test (*p* < 0.05).

**Table 3 microorganisms-12-00251-t003:** OTU number, Chao1 indices, and Shannon diversity indices of fungi and bacteria from different fields.

Sample Location	Sample Trait	Fungi			Bacteria		
		OTU Number	Chao1	Shannon	OTU Number	Chao1	Shannon
Licang, Shandong Province	Healthy	437 ± 13.77 cd	697 ± 56.21 c	4.30 ± 0.09 bc	3237 ± 57.12 b	4204 ± 137.42 b	10.07 ± 0.07 a
	Diseased	361 ± 71.30 d	609 ± 156.53 c	2.39 ± 0.09 d	2860 ± 45.15 c	3908 ± 43.01 b	9.63 ± 0.11 c
Youxian, Sichuan Province	Healthy	852 ± 48.24 a	1235± 49.45 a	5.89 ± 0.24 a	3409 ± 38.58 a	4703 ± 146.66 a	9.99 ± 0.01 b
	Diseased	725 ± 56.10 ab	1136 ± 44.29 ab	5.05 ± 0.23 ab	2703 ± 32.07 c	3954 ± 85.93 b	9.31 ± 0.01 d
Lichuan, Hubei Province	Healthy	574 ± 1.63 bc	809 ± 19.93 bc	4.53 ± 0.26 bc	1918 ± 48.19 f	2690 ± 75.72 d	8.84 ± 0.16 e
	Diseased	483 ± 12.23 bc	775 ± 27.53 bc	3.99 ± 0.45 c	2307 ± 45.11 d	3117 ± 82.60 c	9.24 ± 0.04 dc
Jianshi, Hubei Province	Healthy	641 ± 173.36 ab	938 ± 209.11 ab	4.82 ± 0.46 bc	2124 ± 28.16 e	2968 ± 143.29 cd	9.11 ± 0.09 dc
	Diseased	630 ± 68.78 ab	944 ± 119.06 ab	4.71 ± 0.12 bc	2271 ± 71.98 de	3265 ± 67.68 c	8.96 ± 0.09 de

All data are presented as the mean ± standard deviation (*n* = 3). Values followed by different letters indicate statistically significant differences according to Tukey’s test (*p* < 0.05).

**Table 4 microorganisms-12-00251-t004:** OTU number, Chao1 indices, and Shannon diversity indices of the healthy and diseased soil groups.

	Sample Trait	OTU Number	Chao1	Shannon
Fungi	Healthy	626 ± 174.66 a	920 ± 229.74 a	4.88 ± 0.67 a
	Diseased	550 ± 150.48 a	866 ± 220.62 a	4.03 ± 1.06 b
Bacteria	Healthy	2672 ± 659.28 a	3641 ± 847.00 a	9.50 ± 0.55 a
	Diseased	2535 ± 257.83 a	3560 ± 380.82 a	9.29 ± 0.25 a

All data are presented as the mean ± standard deviation (*n* = 12). Values followed by different letters indicate statistically significant differences according to Tukey’s test (*p* < 0.05).

**Table 5 microorganisms-12-00251-t005:** Pearson’s correlation between incidence of clubroot disease and chemical properties.

	pH	SOM	TON	AN	AP	AK	CEC	EC	Ca	B
Pearson	−0.49	0.421	0.577	0.565	−0.302	0.43	0.983	−0.629	0.456	0.257
*p*-value	0.51	0.579	0.423	0.435	0.698	0.57	0.017 *	0.371	0.544	0.743

* Significant difference at *p* < 0.05.

**Table 6 microorganisms-12-00251-t006:** Pearson’s correlation between the abundance of *P. brassicae* and the relative abundance of the 15 most abundant bacterial and fungal genera.

Fungal Genera	Pearson	*p*-Value	Bacterial Genera	Pearson	*p* Value
*Mortierella*	−0.006	0.978	*Sphingomonas*	−0.245	0.249
*Fusarium*	0.178	0.407	*uncultured_Acidobacteria_bacterium*	−0.098	0.649
*Verticillium*	0.176	0.41	*Gemmatimonas*	−0.323	0.123
*Plectosphaerella*	−0.157	0.465	*Rhodanobacter*	−0.097	0.652
*Penicillium*	−0.022	0.919	*Burkholderia_Caballeronia_Paraburkholderia*	−0.389	0.06
*Chaetomium*	−0.507	0.011 *	*Bradyrhizobium*	−0.236	0.267
*Sordaria*	−0.256	0.227	*Streptomyces*	0.172	0.423
*Botryotrichum*	−0.407	0.049 *	*Nocardioides*	−0.002	0.991
*Saitozyma*	−0.077	0.719	*Pseudolabrys*	−0.240	0.258
*Coniochaeta*	−0.248	0.243	*Bryobacter*	−0.078	0.719
*Solicoccozyma*	0.009	0.966	*Acidibacter*	−0.113	0.598
*Humicola*	−0.250	0.238	*Bacillus*	0.527	0.008 **
*Acremonium*	−0.453	0.026 *	*Ramlibacter*	−0.358	0.086
*Pseudoascochyta*	−0.004	0.987	*Haliangium*	−0.264	0.213
*Trichoderma*	0.111	0.605	*Candidatus_Solibacter*	−0.150	0.485

* Significant difference at *p* < 0.05. ** Significant difference at *p* < 0.01.

**Table 7 microorganisms-12-00251-t007:** Topological indices of each network.

	Fungal Networks		Bacterial Networks	
	Healthy Soils	Diseased Soils	Healthy Soils	Diseased Soils
Number of positive links	2497	2238	4721	3852
Number of negative links	1160	742	4068	2898
Positive link/negative link ratio	2.15	3.02	1.16	1.33
Network density	0.19	0.15	0.44	0.34
Clustering coefficient	0.57	0.53	0.75	0.68
Modularity	0.39	0.40	0.18	0.31

## Data Availability

The sequencing data are available upon request.

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
