# Peer review of "Deciphering Differences in Microbial Community Diversity between Clubroot-Diseased and Healthy Soils"

_microorganisms, 2024, doi:10.3390/microorganisms12020251_

Round 1

Reviewer 1 Report

Comments and Suggestions for Authors

The authors have done a pretty thorough job analyzing a large number of samples, the statistical sampling is done quite competently.

A few minor remarks:

1. The "Discussion" section is written rather vaguely. Quite a large part is devoted to the analysis of correlation networks, but it is not clear what specific result is obtained for the main task of the paper (comparison of clubroot diseased and healthy soil). If you have results on Plasmodiophora brassicae and correlation matrices, you could look for taxa (OTU/ASV or higher) that are positively or negatively correlated with P. brassicae across samples and may somehow influence the development of clubroot disease.

2. Also, when analyzing diversity along with clustering, it would be interesting to look at ordination (beta diversity, PCoA or NMDS), at least in sapplets, to make sure that sampling is correct and that there are no outliers.

3. A general note for the future - I would recommend considering dropping OTU in favor of ASV. And if you're using QIIME, upgrade to version 2 or higher, or use dada2 as a standalone tool.

Author Response

Dear editor and reviewer:

Thank you very much for your letter and for the Reviewers' comments concerning our manuscript entitled “Deciphering differences in microbial community diversity between clubroot-diseased and healthy soils” (Manuscript ID: microorganisms-2819078).

Those comments are all valuable and very helpful for revising and improving our paper. We have studied comments carefully and have made corrections which we hope meet with approval. Revised portions are marked using a word processing program in the paper. The main corrections in the paper and the responses to the Editor and Reviewer's comments are as follows:

Comment 1: The "Discussion" section is written rather vaguely. Quite a large part is devoted to the analysis of correlation networks, but it is not clear what specific result is obtained for the main task of the paper (comparison of clubroot diseased and healthy soil). If you have results on Plasmodiophora brassicae and correlation matrices, you could look for taxa (OTU/ASV or higher) that are positively or negatively correlated with P. brassicae across samples and may somehow influence the development of clubroot disease.

Response: The specific results of this study have been added to the "Discussion" section. To look for taxa that are positively or negatively correlated with P. brassicae across samples, Pearson’s correlations between the abundance of P. brassicae and taxa were performed using SPSS software. We found that the abundance of P. brassicae was significantly negatively correlated with Chaetomium, Botryotrichum, and Acremonium (Table 6). We speculate that these species may play an important role in reducing the development of clubroot disease.

Comment 2: Also, when analyzing diversity along with clustering, it would be interesting to look at ordination (beta diversity, PCoA or NMDS), at least in sapplets, to make sure that sampling is correct and that there are no outliers.

Response: Thank you for your suggestions. We have looked at ordination and the results indicated that sampling is correct and that there are no outliers.

Comment 3: A general note for the future - I would recommend considering dropping OTU in favor of ASV. And if you're using QIIME, upgrade to version 2 or higher, or use dada2 as a standalone tool.

Response: Thank you for your suggestions. We will consider using ASV to investigate soil community diversity in future studies and a higher version of QIIME or dada2 will be used as a standalone tool to analyze data.

We appreciate for editors' and reviewers’ warm work earnestly, and hope that the correction will meet with approval. 

Once again, thank you very much for your comments and suggestions. 

Yours sincerely, 

Huajun Kang

Reviewer 2 Report

Comments and Suggestions for Authors

Overall the manuscript is well presented, and based on a solid data collection. That said, I suggest the data analysis has some problems. For example, you have four fields from two different soil types, each rhizospheric soil collected from diseased or healthy plants. However according to Figure 1, your data analysis just compares the eight samples one against the other without taking any of these factors into consideration. This leads to a major loss of information, from my point of view. That said, the text presenting the results of Figure 1 does indicate an overall difference between diseased and healthy soils, which is not present in the Figure. This problem is repeated in Tables 1 and 2, for example, making it much harder for the reader to discern overall patterns across the samples.

Your data presentation in Table 4 seems to indicate (although the methods section does not make it as clear as it should) that the networks were examined for all four samples of diseased or healthy soils, since there is no indication of variability, just single values for each characteristic. If that was indeed the case, I suggest you evaluate conducting a separate analysis for each soil sample, and then conducting a statistical evaluation (at least for average and some variability measure) for healthy and diseased soils, to allow a proper comparison between them.

Overall, your paper would be much enhanced with a better description of the statistical analysis in the methods section. For example, you included a methodological description in the legend of Figure 6 to indicate how you calculated the measures it presents. 

While correlation is far from being causation, you indicate some linkages between soil variables and disease incidence, which were not properly tested. For instance, you indicate (based on the literature) that soil pH is related to disease incidence, but do not correlate the occurrence of the gene and soil pH, although a simple linear correlation could do that very easily. This is repeated for other variables, and could be easily corrected with some very simple data analysis.

In the same line, you discuss some bacterial and fungal diversity measures and linkages, but although you have four proper replicates of each soil condition, you do not conduct (or at least do not show) any statistical analysis which could attach P values to the conclusions, and so much strengthen them.

Overall, I think this is a good contribution to the field, especially with further data analysis. 

Comments on the Quality of English Language

I do not have any language restrictions to this paper, although I suggest a thorough review after any corrections are made.

Author Response

Dear editor and reviewer:

Thank you very much for your letter and for the Reviewers' comments concerning our manuscript entitled “Deciphering differences in microbial community diversity between clubroot-diseased and healthy soils” (Manuscript ID: microorganisms-2819078).

Those comments are all valuable and very helpful for revising and improving our paper. We have studied comments carefully and have made corrections which we hope meet with approval. Revised portions are marked using a word processing program in the paper. The main corrections in the paper and the responses to the Editor and Reviewer's comments are as follows:

Comment 1: Overall the manuscript is well presented, and based on a solid data collection. That said, I suggest the data analysis has some problems. For example, you have four fields from two different soil types, each rhizospheric soil collected from diseased or healthy plants. However, according to Figure 1, your data analysis just compares the eight samples one against the other without taking any of these factors into consideration. This leads to a major loss of information, from my point of view. That said, the text presenting the results of Figure 1 does indicate an overall difference between diseased and healthy soils, which is not present in the Figure. This problem is repeated in Tables 1 and 2, for example, making it much harder for the reader to discern overall patterns across the samples.

Response: Thank you for your suggestions. We have reanalyzed the data associated with Figure 1 and Table 2. The modification information is as follows.

The abundance of P. brassicae was analyzed in all soil samples collected from four regions (Figure 1). Results showed that the population of P. brassicae ranged from 3.13 Log CFU/g to 8.04 Log CFU/g in the soil samples. Both the smallest population of P. brassicae (3.13 Log CFU/g) and the largest (8.04 Log CFU/g) were detected in the soil samples collected from Youxian. Quantitative analysis showed that the colonization of P. brassicae was less than 3.92 Log CFU/g in the healthy soil samples from the four different fields, but more than 5.28 Log CFU/g of P. brassicae was detected in diseased soils. The P. brassicae population in the healthy soil samples collected from Licang (3.69 Log CFU/g), Lichuan (3.92 Log CFU/g), and Jianshi (3.80 Log CFU/g) was significantly higher than that collected from Youxian (3.13 Log CFU/g). However, the P. brassicae population in the diseased soil samples collected from Youxian was significantly higher than that collected from the other three fields. Among the four diseased soil samples, the lowest level (5.28 Log CFU/g) of resting spore infestation was detected in Lichuan. Within the same soil type, the abundance of P. brassicae in the healthy soils was significantly lower than that of the diseased soils. For the same field, the P. brassicae population in the diseased soil samples was significantly (Tukey’s test, P < 0.05) higher than in the healthy soil samples.

 The chemical properties of the soils are presented in Table 2. The pH, AN, AP, AK, and Ca contents in healthy soil samples were significantly different among the four regions. For diseased soil samples, AK, AP, and Ca levels were significantly different among the four regions. For each field or each of the two soil types, compared with the healthy soils, the diseased soils had a higher AK content, but with no significant difference. There were significant differences in Ca levels between the healthy and diseased soils in the same field or each soil type. The pH of the healthy and diseased soils was significantly different in Youxian, Lichuan, and Licang, respectively, but did not show a consistent tendency in different fields. The CEC and B contents showed no difference between the healthy and diseased soils in the same field. However, other chemical properties (SOM, TN, AN, AP, and EC) showed different trends between the healthy and diseased soils in different regions.

Comment 2: Your data presentation in Table 4 seems to indicate (although the methods section does not make it as clear as it should) that the networks were examined for all four samples of diseased or healthy soils, since there is no indication of variability, just single values for each characteristic. If that was indeed the case, I suggest you evaluate conducting a separate analysis for each soil sample, and then conducting a statistical evaluation (at least for average and some variability measure) for healthy and diseased soils, to allow a proper comparison between them.

Response: The construction of each co-occurrence network requires more than five replicates of samples. There are only three replicates of healthy or diseased soil samples per field, so it is not possible to construct a network for each sample alone. We have modified the construction method of the co-occurrence network in the methods section. Thank you for pointing out this problem in the manuscript. More repetitions will be contained in the future studies.

Comment 3: Overall, your paper would be much enhanced with a better description of the statistical analysis in the methods section. For example, you included a methodological description in the legend of Figure 6 to indicate how you calculated the measures it presents. 

Response: Thank you for your suggestions. We have added a more detailed description of the statistical analysis in the methods section.

Comment 4: While correlation is far from being causation, you indicate some linkages between soil variables and disease incidence, which were not properly tested. For instance, you indicate (based on the literature) that soil pH is related to disease incidence, but do not correlate the occurrence of the gene and soil pH, although a simple linear correlation could do that very easily. This is repeated for other variables, and could be easily corrected with some very simple data analysis.

Response: Thank you for your suggestions. The correlation between soil variables and disease incidence was analyzed using Pearson’s correlation coefficient analysis. Disease incidence was only significantly positively correlated with CEC (Table 5). There was no correlation between disease incidence and other chemical properties.

Comment 5: In the same line, you discuss some bacterial and fungal diversity measures and linkages, but although you have four proper replicates of each soil condition, you do not conduct (or at least do not show) any statistical analysis which could attach P values to the conclusions, and so much strengthen them.

Response: Thank you for your suggestions. The differences in bacterial and fungal diversity between the healthy and diseased soil groups have been statistically analyzed. We found that the fungal Shannon diversity indices in healthy soil groups were significantly (Tukey’s test, P < 0.05) higher than those in diseased soil groups (Table 4).

Comment 5: Overall, I think this is a good contribution to the field, especially with further data analysis.

Response: Thank you very much for your letter. These comments are all valuable and very helpful for revising and improving our paper. We have further analyzed the data according to your comments.

We appreciate for editors' and reviewers’ warm work earnestly, and hope that the correction will meet with approval. 

Once again, thank you very much for your comments and suggestions. 

Yours sincerely, 

Huajun Kang